# The Regulatory Role of *Myomaker* in the Muscle Growth of the Chinese Perch (*Siniperca chuatsi*)

**DOI:** 10.3390/ani14172448

**Published:** 2024-08-23

**Authors:** Wei Zeng, Yangyang Meng, Lingtao Nie, Congyi Cheng, Zexia Gao, Lusha Liu, Xin Zhu, Wuying Chu

**Affiliations:** 1Hunan Provincial Key Laboratory of Nutrition and Quality Control of Aquatic Animals, Hunan Engineering Technology Research Center for Amphibian and Reptile Resource Protection and Product Processing, College of Biological and Chemical Engineering, Changsha University, Changsha 410022, China; zengwei_2014@163.com (W.Z.); mengyangyang163938@163.com (Y.M.); m15842923934@163.com (C.C.); 2Key Laboratory of Agricultural Animal Genetics, Breeding and Reproduction of Ministry of Education, Wuhan 430070, China; gaozx@mail.hzau.edu.cn (Z.G.); liuls@mail.hzau.edu.cn (L.L.); 3Key Laboratory of Freshwater Animal Breeding, Ministry of Agriculture and Rural Affairs, College of Fisheries, Huazhong Agricultural University, Wuhan 430070, China

**Keywords:** *Myomaker*, skeletal muscle, myoblast fusion, *Siniperca chuatsi*

## Abstract

**Simple Summary:**

*Myomaker* has been reported to play an important role in regulating myoblast fusion. However, the role of *Myomaker* gene in skeletal muscle growth in economic fish during post-hatching stage is unclear. This study showed that the growth of Chinese perch was significantly decreased when *Myomaker* was inhibited by *Myomaker*-siRNA. Furthermore, both the diameter of muscle fibers and the number of nuclei in single muscle fibers were significantly reduced in the *Myomaker*-siRNA group, whereas, there was no significant difference in the number of proliferating cells between the control and *Myomaker*-siRNA groups. Together, these findings indicate that *Myomaker* may promote the hypertrophy of muscle fibers and growth of fast muscle in Chinese perch by promoting myoblast fusion.

**Abstract:**

The fusion of myoblasts is a crucial stage in the growth and development of skeletal muscle. *Myomaker* is an important myoblast fusion factor that plays a crucial role in regulating myoblast fusion. However, the function of *Myomaker* in economic fish during posthatching has been poorly studied. In this study, we found that the expression of *Myomaker* in the fast muscle of Chinese perch (*Siniperca chuatsi*) was higher than that in other tissues. To determine the function of *Myomaker* in fast muscle, *Myomaker*-siRNA was used to knockdown *Myomaker* in Chinese perch and the effect on muscle growth was determined. The results showed that the growth of Chinese perch was significantly decreased in the *Myomaker*-siRNA group. Furthermore, both the diameter of muscle fibers and the number of nuclei in single muscle fibers were significantly reduced in the *Myomaker*-siRNA group, whereas there was no significant difference in the number of BrdU-positive cells (proliferating cells) between the control and the *Myomaker*-siRNA groups. Together, these findings indicate that *Myomaker* may regulate growth of fast muscle in Chinese perch juveniles by promoting myoblast fusion rather than proliferation.

## 1. Introduction

Skeletal muscle represents a significant proportion of the body weight of fish, comprising between 40% and 60% of the total body weight [1]. Fish skeletal muscle is composed of anatomically and functionally distinct fiber layers, namely fast and slow muscle. Fast muscle represents the primary component of skeletal muscle, comprising over 80% of the muscle, and is situated in the deep layer of the myotome. In contrast, slow muscles constitute a smaller proportion of skeletal muscle and are a superficial longitudinal band located beneath the skin [2,3]. After birth, the number of muscle fibers generally does not increase in mammals—the muscle fiber hypertrophy is maintained primarily by the muscle fiber lengthening and thickening [4]. In contrast to mammals, the muscle growth in many fish species during the posthatching period is the combined effect of an increase in the number of muscle fibers (hyperplasia) and an increase in the size of pre-existing fibers (hypertrophy) [5]. Hyperplasia occurs through the activation of quiescent satellite cells that differentiate to form mononuclear myocytes. Muscle fiber hypertrophy is achieved by two main methods: one is regulated by the stimulation of protein synthesis as well as the activation of ribosomal RNA and muscle-specific gene expression, and the other is the fusion of myoblasts to form new muscle fiber [6,7]. The number of muscle fibers in fish continues to increase during posthatching, indicating that muscle fiber hypertrophy in fish is the result of a combination of two approaches [4,8,9]. Fish growth involves hypertrophy and hyperplasia of skeletal muscle, which is crucial for their muscle development [10]. Currently, there is limited knowledge on the growth of muscle fibers in fish during the posthatching period, particularly the growth caused by the mutual fusion of myoblasts.

Myoblast fusion is a complex process that mainly involves a series of processes including migration, recognition, adhesion, membrane alignment, cytoskeletal rearrangement, and fusion pore formation [11]. Myoblast fusion occurs in three steps: the first step is recognition and adhesion between the muscle cells; the second step is increasing proximity of cell membranes; finally, lipid bilayers must be disrupted to facilitate fusion pore formation and allow the exchange of cytoplasmic material and, finally, fusion into a single cell [12]. Among them, the cell adhesion molecules *Jamb* and *Jamc*, the cell migration-associated molecule *Cdc42*, and the cytoskeleton-associated membrane proteins *Rac1* and *Stability-2* have been reported to be involved in the fusion process of myoblasts [11,13,14,15], but muscle-specific regulatory proteins have rarely been reported to be involved in this process.

The myoblast fusion factor *Myomaker* was first discovered in 2013 [16], and its discovery provided important insights into the molecular and cellular mechanisms of myoblast fusion. *Myomaker* is expressed on the surface of the cell membrane of myoblasts and plays a major role in the initiation of fusion and the formation of hemifusion intermediates, which are essential for myoblast fusion [17]. *Myomaker* also plays an important role in the repair process after muscle injury in adult animals, and its role in myoblast fusion and muscle repair has been demonstrated in avian species such as chickens (*Gallus domestiaus*) [18,19]. Knockout of *Myomaker* in Zebrafish partially survived, but *Myomaker* knockout zebrafish were only one-third the weight of wild-type siblings, and the muscle fibers were single-nucleus muscle fiber with a much smaller diameter [20]. Although the *Myomaker* function in zebrafish has been studied, the zebrafish is not an ideal model for studying muscle growth after hatching because it is a model organism of determinate growth, reaching a final size of only 3–5 cm [21]. In contrast, the growth of most aquaculture fish is indeterminate, and individual growth is still achieved during the posthatching stage by the proliferation of myoblast (hyperplasia) and an increase in the size of existing muscle fibers (hypertrophy) [6,22]. Consequently, investigating the function of *Myomaker* in the postembryonic skeletal muscle development of economic fish represents a significant reference point for the advancement of fish breeding and culture.

Chinese perch (*Siniperca chuatsi*) is one of the important and valuable economic freshwater fish in China [23]. Due to its advantages of no intermuscular bone, delicious taste, and rich nutrition, Chinese perch is increasingly preferred by consumers, leading to a rising market demand [24]. In this study, we analyzed the sequence characteristics of Chinese perch *Myomaker* and detected the expression pattern of *Myomaker* in different tissues of Chinese perch. To explore its role in muscle growth in Chinese perch juveniles, *Myomaker* expression was inhibited using *Myomaker*-siRNA. This study may provide more information for improving our knowledge on fish skeletal muscle growth and a theoretical basis for the breeding and culture of Chinese perch.

## 2. Materials and Methods

### 2.1. Experimental Animals

The Chinese perch utilized in the tissue expression experiments (five-month-old, weighing 210 ± 10 g) and the siRNA interference experiments (one-month-old, weighing 2.3 ± 0.3 g) were obtained from the Hunan Fisheries Science Institute (Changsha, China). The Chinese perch were reared in recirculating aquaria with an incubation water temperature of 24 ± 1 °C, a dissolved oxygen level of 8 ± 0.2 mg/L, a pH range of 7.4–7.7, and were fed equal amounts of live *Megalobrama amblycephala* juveniles twice daily (8:00 am and 6:00 pm). This study followed the guidelines approved by the Institutional Animal Care and Use Committee of Changsha University (Changsha, China), and all operations were performed to ensure the minimum pain of experimental fish.

### 2.2. Design and Preparation of siRNAs

The complete CDS sequence of Chinese perch *Myomaker* was obtained from the Chinese perch genome database (http://genomes.igb-berlin.de/Siniperca/ (accessed on 12 March 2023)). *Myomaker* silencing was performed by cholesterol modified siRNA targeting *Myomaker* (*Myomaker*-siRNA) (sense 5′-GCAGCUGAGAGCAGUGUAUTT-3′, antisense 5′-AUACACUGCUCUCAGCUGCTT-3′), and nonsense siRNA (sense 5′-UUCUCCGAACGUGUCACGUTT-3′, antisense 5′ACGUGACACGUUCGGAGAATT-3′) was used as control. A single cholesterol moiety was linked to the 3′ end of the passenger strand. The *Myomaker*-siRNAs were designed from mRNA sequence of Chinese perch *Myomaker* and synthesized by GenePharma (Shanghai, China).

### 2.3. Tissue Sampling and Myomaker-siRNA Inhibition Assay

Chinese perch were anesthetized with tricaine methanesulfonate (MS-222, 0.15 g/L), and tissue specimens of liver, spleen, kidney, brain, gut, dorsal fast muscle, and slow muscle were collected from six Chinese perch on ice. The slow muscle is characterized by its red color and located along the lateral sides of the fish, and the fast muscle is white and located in the deep layer of the dorsal myotome. All the tissue samples were stored at −80 °C. The Chinese perch used for *Myomaker*-siRNA interference experiments were randomly divided into control and *Myomaker*-siRNA groups, with six fish in each group. The initial length and weight of the Chinese perch were recorded. The interference times and doses of *Myomaker*-siRNA were determined based on previous reports [25,26]. The fish in the *Myomaker*-siRNA group received 20 μM *Myomaker*-siRNA injections by dorsal muscle injection at a dosage of 2 mg/kg every seven days for 21 days. The control group was injected at the same concentration and dose of nonsensical siRNA. After 21 days, 10 mg/kg BrdU was injected into the dorsal muscle 6 h before sampling. The Chinese perch were anesthetized with tricaine methanesulfonate (MS-222, 0.15 g/L) and the body length and weight were measured before sampling. Using a scalpel and forceps to gently separate the dorsal skin from the underlying fast muscle, the fast muscle below the dorsal fin was collected, and the sampling area of the fast muscle was shown in Appendix A. The samples of the dorsal fast muscle were split into two sections; one part was preserved at −80 °C for RNA extraction, and the other was fixed with 4% paraformaldehyde overnight at 4 °C for immunofluorescence analysis and histological section.

### 2.4. Bioinformatics Analysis of Myomaker

Sequences of Myomaker proteins were extracted from the NCBI database for Chinese perch and other species. Sequence alignments were conducted using DNAMAN, and the physical and chemical characteristics of the proteins were analyzed with Expasy’s ProtParam Proteomics server. In order to predict signal peptides and transmembrane structural domains, SMART (http://smart.embl-heidelberg.de/ (accessed on 4 October 2023)) and TMHMM Server v. 2.0 (http://www.cbs.dtu.dk/services/TMHMM-2.0/ (accessed on 5 October 2023)) were employed.

### 2.5. cDNA Synthesis and Quantitative Real-Time PCR Analysis

Total RNA from all samples was extracted using RNAiso Plus (Takara, Beijing, China), and the concentration and quality of the extracted RNA were detected by ultra-micro spectrophotometer (NanoPhotometer-NP80, Implen, Munich, Germany) and 2% agarose gel electrophoresis. Equal amounts of RNA were reverse transcribed using PrimeScript™ RT reagent Kit with gDNA Eraser (Takara, Beijing, China). Relative transcript levels were measured by quantitative PCR using MonAmp™ SYBR^®^ Green qPCR Mix (Monad, Suzhou, China). The qRT-PCR method was implemented following our previous report [27], with the *Rpl13* gene serving as the reference gene. Statistical variance was tested using independent-samples *t*-test and one-way ANOVA. Additionally, the normal distribution of the data was analyzed using Shapiro–Wilk analysis. The relative expression level of the target mRNA was determined through R = 2^−ΔΔCt^ calculation [28]. Primers for the qRT-PCR assays were designed by Primer Premier 5.0 software, and the sequences were shown in Table 1.

### 2.6. Histological Section and Immunofluorescence Analysis

The samples were dehydrated in a gradient ethanol solution, xylene clear, paraffin embedded, and the muscle tissue was cut into 6 μm using a Leica SM2010R slicer. H&E staining was conducted utilizing the Hematoxylin and Eosin Staining Kit (Solarbio, Beijing, China) in accordance with the manufacturer’s guidelines. The samples were subsequently observed under an inverted microscope (DMI3000B, Leica, Wetzlar, Germany), and standard photographs were analyzed with the Image-Pro Plus 6.0 software (Media Cybernetics, Bethesda, MD, USA). A random field of view was selected for each section in ImageJ (version 1.46 r), and then each of the muscle-fiber cross-sectional areas in 1 mm^2^ field of view was counted. A total of six histological section were counted for each of the controls and *Myomaker*-siRNA groups. The diameters of the muscle fibers were calculated by approximating the cross-section of the muscle fibers as a circle. For immunofluorescence detection, paraffin sections were deparaffinized to water, and antigen repair with Antigen Repair Solution (G1202, servicebio, Wuhan, China). The sections were blocked with 10% goat serum for 30 min and then incubated overnight with anti-BrdU (Servicebio, GB12051, Wuhan, China). Subsequently, sections were incubated with fluorophore-conjugated secondary antibodies (GB21301, servicebio, Wuhan, China), and nuclei co-staining with DAPI (G1012, servicebio, Wuhan, China). Images were acquired with a laser scanning confocal microscope (LSM710, Zeiss, Oberkochen, Germany).

### 2.7. Isolation and DAPI Staining of Single Muscle Fibers of Chinese Perch

Fast muscle of the under-dorsal fin of Chinese perch was taken for single muscle fiber DAPI staining. Ten muscle fibers were sampled from each of the three randomly selected fish in each group. The single muscle fibers were separated according to the method of Shi [18], and the dissected muscle fibers were placed on slides. Staining was performed with 1 μg/mL of DAPI and washed three times with PBS for five minutes each after staining, and a fluorescence microscope (DMI3000B, Leica, Wetzlar, Germany) was used to take pictures.

### 2.8. Statistical Analysis

The data were subjected to statistical analysis using the SPSS 19.0 software package. All data are presented as mean ± SEM. The statistical variance was tested using independent-samples *t*-test and one-way ANOVA, and *p* values < 0.05 was considered to indicate a statistically significant difference between the two groups. Furthermore, the data were subjected to a Shapiro–Wilk analysis to ascertain their normal distribution.

## 3. Results

### 3.1. Amino Acid Sequence Comparison and Homology Analysis of the Myomaker Protein

The full-length Chinese perch *Myomaker* gene encodes a protein of 285 amino acids. The molecular weight, molecular formula, instability index (II), and theoretical isoelectric point (pI) of the Chinese perch *Myomaker*-encoded protein were 32.032 kDa, C_1477_H_2261_N_363_O_391_S_21_, 38.77 and 9.43, respectively. A DUF3522 superfamily (starting at position 3 amino acids and ending at position 185 amino acids) and a low complexity region (starting at position 225 amino acids and ending at position 240 amino acids) were detected using SMART. The Myomaker protein in Chinese perch exhibits sequence identities of 72% and 88% with human and zebrafish Myomaker proteins, respectively. The protein sequences of Myomaker are highly conserved during the process of evolution (Figure 1A). Myomaker protein is predicted to have five transmembrane domains and a signal peptide at the N-terminus. The Chinese perch Myomaker contains five transmembrane domains, the same number of transmembrane domains as *Micropterus salmoides*, *Sander lucioperca*, and *Xenopus laevis* (Figure 1B,C). The majority of the protein is located within the plasma membrane, with small extracellular domains (Figure 1B).

### 3.2. Tissue Specific Expression of Myomaker

To examine the expression level of *Myomaker* in different tissues, different tissue samples from Chinese perch juveniles were collected and analyzed using quantitative real-time PCR assays. *Myomaker* mRNA were detected in all tested tissues, but *Myomaker* was significantly expressed in skeletal muscle and was higher in fast muscle than in slow muscle, while there were no significant differences between other tissues (Figure 2).

### 3.3. Interference with Myomaker Inhibits Growth of Chinese Perch Juvenile

To verify the knockdown efficiency of *Myomaker*-siRNA, the relative expression of *Myomaker* was detected in control and *Myomaker*-siRNA group by RT-qPCR. The results showed that *Myomaker* expression was reduced by the *Myomaker*-siRNA (Figure 3C), and the difference was statistically significant compared with control (*p* < 0.05). After 21 days of *Myomaker*-siRNA interference, the body length in control increased by 1.6 ± 0.26 cm, while the *Myomaker*-siRNA group increased by 1.0 ± 0.3 cm (Table 2, Figure 3A,B,D). The body weight in control increased by 4.1 ± 0.31 g, while the *Myomaker*-siRNA group increased by 1.9 ± 0.26 g (Table 2, Figure 3E). There was a significant difference in body length and body weight gain between the control and *Myomaker*-siRNA groups. The results indicated that the growth of Chinese perch juveniles was significantly inhibited after interfering with *Myomaker* expression.

### 3.4. Histological Section Analysis

The study indicates a significant reduction in the diameter of muscle fibers in the *Myomaker*-siRNA injection group compared to the control group (Figure 4A–C). The diameter of muscle fibers in the control was mainly concentrated in 40–50 µm (Figure 4D), and the diameter of muscle fibers in the *Myomaker*-siRNA group was mainly concentrated in 20–30 µm (Figure 4E). The results showed that the expression of *Myomaker* played an important role in the thickening of muscle fibers in Chinese perch.

### 3.5. The Nuclei Number Analysis of Single Muscle Fibers

Isolated single muscle fibers were stained for cell nuclei using DAPI. The results showed that the number of nuclei in single muscle fibers was significantly reduced in the *Myomaker*-siRNA group compared to the control group (Figure 5). It is suggested that interfering with *Myomaker* expression inhibited the fusion of muscle fiber, thus leading to a reduction in the nuclei of muscle fibers in the *Myomaker*-siRNA group.

### 3.6. Effect of Inhibiting Myomaker on Myoblast Proliferation and Differentiation

Immunofluorescence results showed that there was no significant difference in the number of myoblasts proliferating between the control and *Myomaker*-siRNA groups (Figure 6A–G), and RT-qPCR results showed that there was no significant difference in the expression of *Pax7*, *MyoD*, and *MyoG* between the control and *Myomaker*-siRNA groups (Figure 6H–J). The results indicate that *Myomaker* promoting the hypertrophy of muscle fibers is not achieved by regulating the proliferation and differentiation of myoblasts.

## 4. Discussion

The fusion of myoblasts to form myotubes is an important step in skeletal muscle development [29]. Myomaker is a myogenic fusion protein that controls myoblast fusion [16]. Even though the mouse (*Mus musculus*) and zebrafish *Myomaker* genes have been extensively studied, the effect on skeletal muscle fusion during the posthatching period in economic fish has been limited [16]. The sequences of Myomaker protein in Chinese perch were analyzed initially. Furthermore, interference with *Myomaker* expression using *Myomaker*-siRNA confirmed that *Myomaker* is required for myoblast fusion in Chinese perch by detecting proliferation and fusion in skeletal muscle.

Myomaker proteins in tetrapods and zebrafish comprise 221–220 amino acids [20,30]. In contrast, other teleost contained the Myomaker protein comprising over 220 amino acids [30]. In trout (*Oncorhynchus mykiss*), the Myomaker proteins consist of 434 amino acids containing 14 minisatellites consisting of repetitive sequences in addition to the preceding 220 highly consistent amino acids [30]. In this study, bioinformatic analyses of the structural and amino acid sequence homology of the Chinese perch Myomaker protein showed that the Chinese perch *Myomaker* gene encodes 285 amino acids. The first half (1–220 amino acid) of the Chinese perch Myomaker protein sequence was highly similar to that of human and zebrafish, but the Chinese perch Myomaker protein is 65 amino acids longer than the zebrafish orthologues. The extended region at the C-terminus contains a low complexity region starting at position 225 and ending at position 240, which is a region of low diversity consisting of residues (nucleotides or amino acids). A low complexity region comprises low diversity regions of residues (nucleotides or amino acids). Proteins with low complexity regions play a role in biological processes, such as transcription, stress responses, and extracellular structures [31]. The C-terminal extended region of the Myomaker protein is different in different species, and whether differences in the C-terminal extended region affect Myomaker function needs to be further investigated.

In mouse and zebrafish, the Myomaker protein contains seven transmembrane domains, but no signal peptide was found at the N-terminus [20,32]. In contrast, TMHMM Server v. 2.0 predicts that the Chinese perch Myomaker protein contains five transmembrane domains and a signal peptide at the N-terminus. A signal peptide is usually found at the amino terminus of secreted proteins and is further processed by targeting proteins for secretion or specific organelles [33]. The presence of a signal peptide at the N-terminus of the Chinese perch Myomaker protein suggests that Myomaker proteins may require further processing in the organelle before functioning.

In mouse, porcine, and chicken, *Myomaker* was shown to be a myoblast fusion factor that plays an important role in promoting myoblast fusion [16,19,34]. Although the expression pattern of *Myomaker* is similar in mouse, porcine, and chicken, this conservation has been less studied in commercial freshwater fish. In posthatching of yellowfin seabream (*Acanthopagrus latus*), *Myomaker* was widely distributed in all tissues, with the highest levels of *Myomaker* expression detected in fast muscle [35]. The *Myomaker* gene is expressed only in the fast and slow muscle of trout posthatching, with little detectable *Myomaker* expression in other tissues [30]. In this study, *Myomaker* expression was observed to be significant in the skeletal muscles of Chinese perch juvenile. Additionally, the results were consistent with those found in yellowfin seabream and trout [30,35]. However, *Myomaker* expression was not observed in the muscles of adult mice [14]. This may be due to the fact that myoblast proliferation, fusion, and differentiation remain active in fish posthatching [6].

Fish muscle fibers are segmented into different layers, with fast muscles making up 90–95% of the total muscle mass in most fish [5]. In contrast, slow muscles comprise a smaller proportion of muscle mass and typically run parallel to the lateral line [5]. In the present study, the highest levels of *Myomaker* expression were detected in the fast muscle followed by the slow muscle of Chinese perch juveniles. However, *Myomaker* is highly expressed in the fast muscle of embryonic zebrafish, but significantly decreased in the slow muscle [20,36]. It may be because slow muscle fibers are mononuclear at the embryonic stage. However, during the juvenile stage, slow muscles express *Myomaker*, which allows them to fuse and become multinucleated at the posthatching stage [37].

Myomaker is directly involved in the fusion process of myoblasts and participates in membrane hemifusion, which is essential for myoblast fusion to form multinucleated muscle fibers [16,17]. The majority of posthatching research has concentrated on the function of *Myomaker* in the repair process following muscle injury. During mouse embryogenesis, the *Myomaker* gene shows strong expression in the myotomal segment of the somites [16]. Upon finishing the formation of the muscles, the gene is down-regulated with a pattern of expression that closely mirrors that of *MyoD* and *MyoG* [16]. After disruption of *Myomaker* expression, all mice died perinatally due to lack of multinucleated muscle fibers [16]. Studies of *Myomaker* have focused on its effects on myoblast fusion at the embryonic stage and in vitro [16,34,38]. However, myoblast still have the ability to proliferate in fish larvae, and the fusion of myoblasts remains active [6]. Thus, *Myomaker* remains continuously expressed in fish posthatching. Studies have shown that *Myomaker* is required for the fusion and growth of myoblasts in the fast and slow muscles of the zebrafish during posthatching [37]. Knockout of *Myomaker* in zebrafish embryos results in defective myoblast fusion and increased adipocyte infiltration in skeletal muscles [20]. In this study, we interfered with the expression of *Myomaker* in the fast muscle of Chinese perch juveniles by *Myomaker*-siRNA. The results showed a significant reduction in body weight gain in the perch *Myomaker*-siRNA group compared to the control, suggesting that interference with *Myomaker* during the juvenile stage of Chinese perch results in slow growth. Nevertheless, the diameter of muscle fibers and the number of nuclei in single muscle fibers were significantly reduced in juvenile fish following *Myomaker*-siRNA interference compared to control, which is consistent with the finding of Shi et al. [20]. The results of this study suggest that the inhibition of *Myomaker* expression may lead to impaired fusion between myoblasts and inhibited muscle fiber hypertrophy, which in turn may result in impaired growth and development of Chinese perch juveniles.

The expression of muscle-specific transcripts is not affected in *Myomaker* knockout mice, indicating that *Myomaker* expression is not associated with myoblast differentiation and proliferation [16,39]. Nevertheless, recent studies in mouse have shown that *Myomaker* expression was down-regulated in C2C12 myoblasts and cell differentiation was inhibited [40]. The expression of *Myomaker* was down-regulated in porcine adult myoblasts, and the expression levels of *MyHC* and *MyoG* were significantly reduced [34]. There are differences in the effects on muscle differentiation and proliferation depending on the method used to down-regulate *Myomaker* expression. Immunofluorescence and RT-qPCR assays were conducted to investigate the impact of *Myomaker* gene expression suppression via *Myomaker*-siRNA interference on myoblast differentiation and proliferation. The findings indicated that there was no significant difference in myoblast proliferation between the control and the *Myomaker*-siRNA groups. Moreover, interfering with *Myomaker* expression has no influence on the expression of *Pax7*, *MyoD*, and *MyoG*. It is shown that interference with *Myomaker* expression by *Myomaker*-siRNA does not affect myoblast proliferation and differentiation.

## 5. Conclusions

In the study, interference with *Myomaker* expression by *Myomaker*-siRNA resulted in a slowing of the growth rate of Chinese perch juveniles, and a decrease in the diameter of muscle fibers and the number of nuclei of single muscle fibers. However, myoblast proliferation is not affected, suggesting that *Myomaker* is involved in skeletal muscle hypertrophy by enhancing myoblast fusion rather than regulating muscle cell proliferation.

## Figures and Tables

**Figure 1 animals-14-02448-f001:**
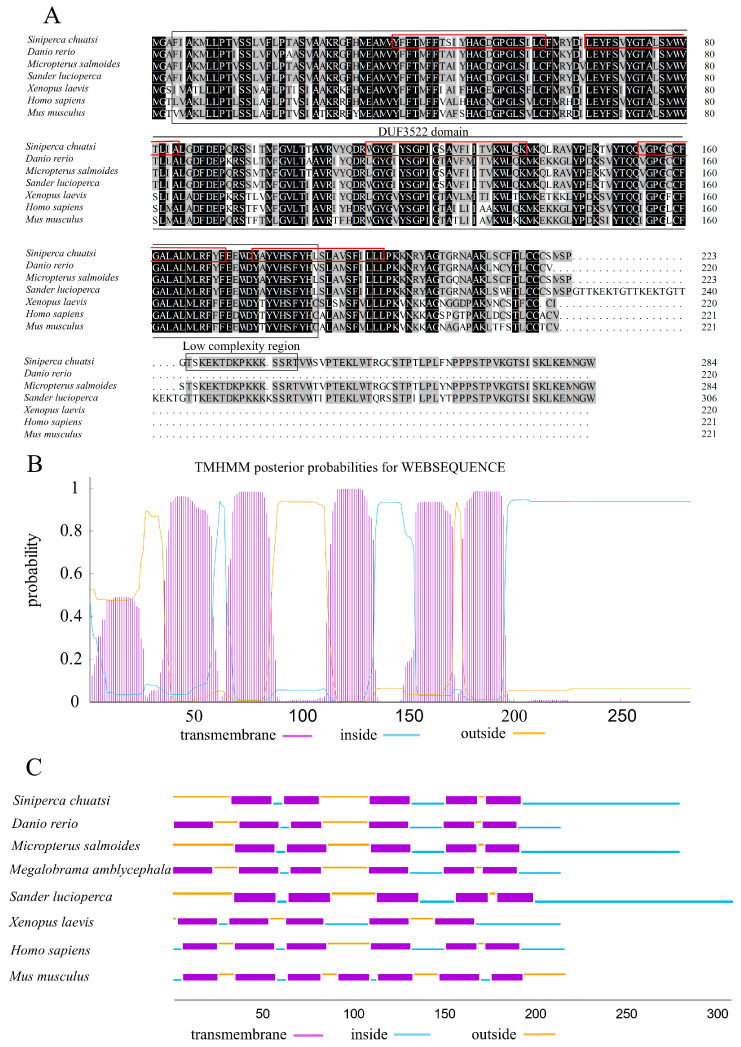
Amino acid sequence comparison and homology analysis of the Myomaker protein. (**A**): Amino acid sequence alignment of Myomaker of Chinese perch and other species. Identical amino acid residues are indicated with a black background and similar amino acid residues are shaded. Predicted protein domains are shown with boxes, and the red boxes show the transmembrane domain of the Chinese perch Myomaker. (**B**,**C**): Predictions of transmembrane domains in Chinese perch and other species Myomaker protein using the TMHMM Server v. 2.0. The *x*-axis indicates the number of amino acids. The *y*-axis represents the probability of each amino acid being in the transmembrane region, inside and outside the cell membrane. GeneBank accession numbers: *Danio rerio*, NP_001002088.1; *Micropterus salmoides*, XP_038559766.1; *Megalobrama amblycephala*, XP_048045299.1; *Siniperca chuatsi*, XP_044052078.1; *Sander lucioperca*, XP_031161515.1; *Xenopus laevis*, XP_041428255.1; *Mus musculus*, NP_079652.1; *Homo sapiens*, NP_001073952.1.

**Figure 2 animals-14-02448-f002:**
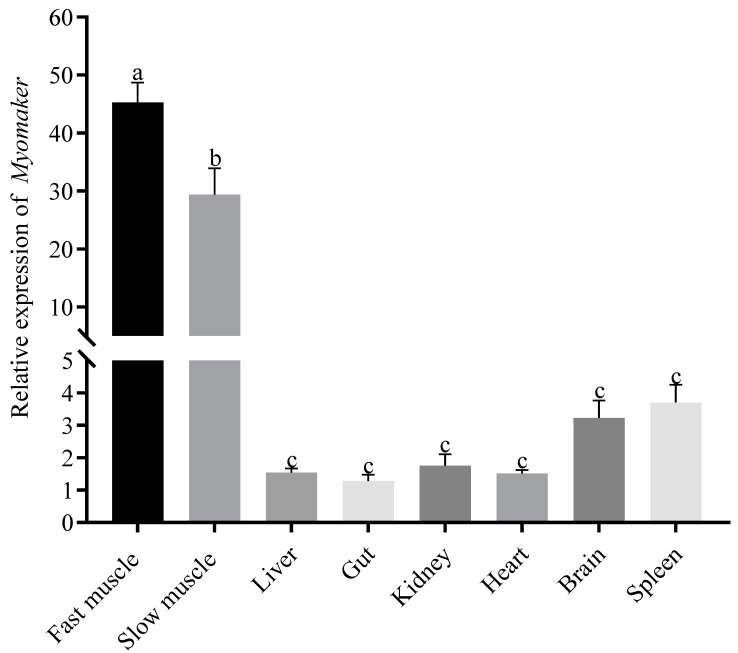
The tissue expression distribution of *Myomaker* in Chinese perch juveniles. The different colored columns represent the relative expression levels of *Myomaker* in different tissues. Values in the figures are the mean ± SEM, *n* = 6. Letters on the error line indicate significance markers, and different letters represent statistical difference between different tissues (*p* < 0.05).

**Figure 3 animals-14-02448-f003:**
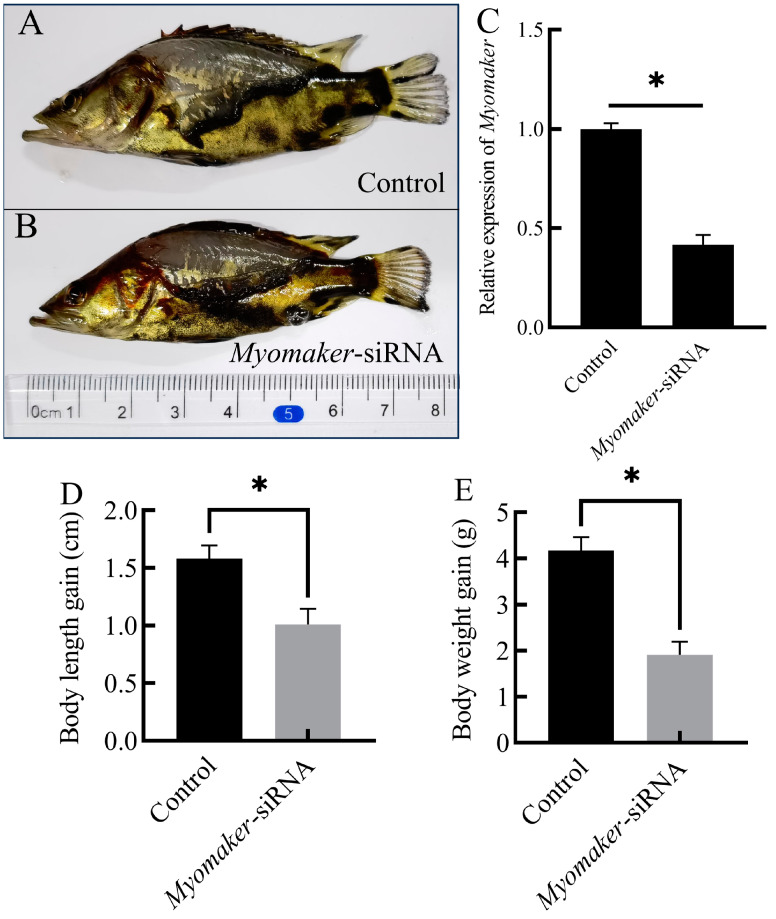
Growth changes in Chinese perch after *Myomaker*-siRNA injection. Photographs of control and *Myomaker*-siRNA after 21 days of *Myomaker*-siRNA interference (**A**,**B**). Relative expression levels of *Myomaker* in Chinese perch fast muscle of nonsense siRNA (Control) and *Myomaker*-siRNA groups (**C**). Comparison of body length and body weight between control and *Myomaker*-siRNA after 21 days of *Myomaker*-siRNA interference (**D**,**E**). Values in the figures are the mean ± SEM, *n* = 6. * Indicates the significant difference in expression between the control and the *Myomaker*-siRNA groups (*p* < 0.05).

**Figure 4 animals-14-02448-f004:**
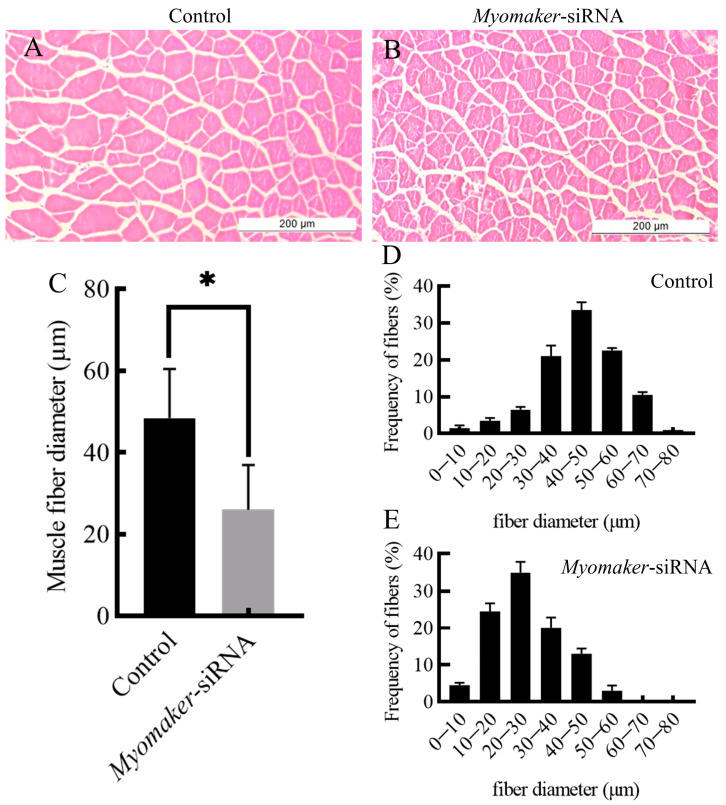
Histological section analysis of skeletal muscle in control and *Myomaker*-siRNA groups. HE staining showing cross sections of fast muscles in control and *Myomaker*-siRNA groups (**A**,**B**). The muscle fibers diameter in control and *Myomaker*-siRNA groups (**C**). Frequency distribution of muscle fiber diameter in control and *Myomaker*-siRNA groups (**D**,**E**). Values in the figures are the mean ± SEM, *n* = 6. * Indicates the significant difference in expression between the control and *Myomaker*-siRNA groups (*p* < 0.05).

**Figure 5 animals-14-02448-f005:**
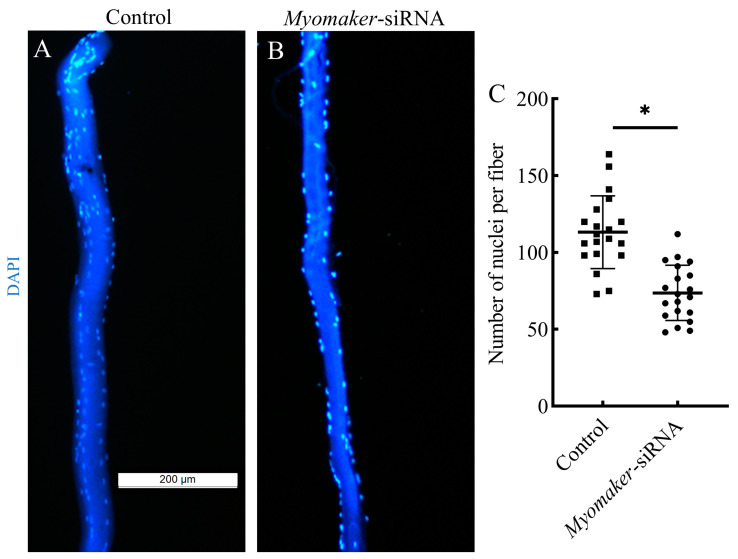
Single fiber analysis of nuclear numbers in skeletal muscles from control and *Myomaker*-siRNA groups. Control (**A**) and *Myomaker*-siRNA (**B**) groups single muscle fibers were isolated were stained by DAPI (blue), and photographed by fluorescence microscopy. Statistical analysis of nuclear numbers in muscle fibers from control and *Myomaker*-siRNA groups (**C**). Values in the figures are the mean ± SEM, *n* = 20. Each square and circle represent the number of single fiber nuclei in the control group and the *Myomaker*-siRNA group, respectively. * Indicates the significant difference between the control and the *Myomaker*-siRNA groups (*p* < 0.05).

**Figure 6 animals-14-02448-f006:**
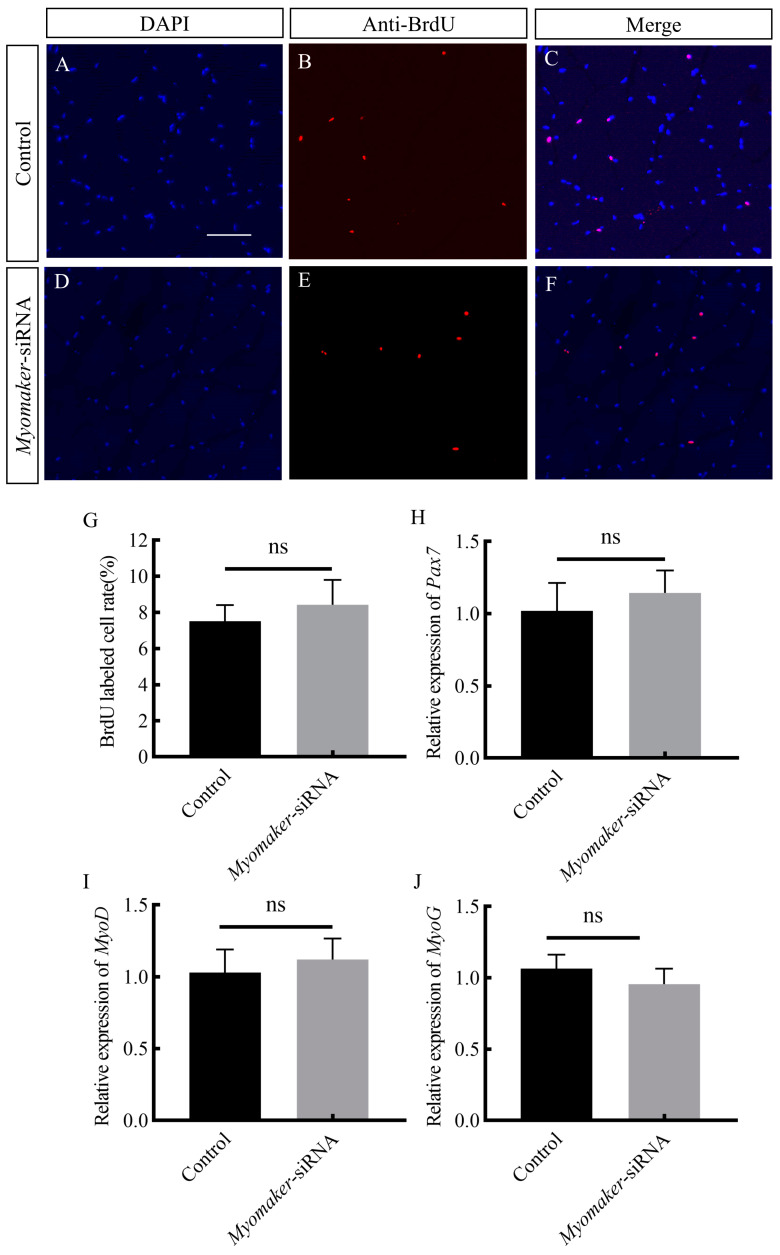
Effects of interfering with *Myomaker* expression on myoblast proliferation and differentiation. Immunofluorescence labeling of the transverse section of muscle fibers in control and *Myomaker*-siRNA groups (**A**–**F**). Nuclei were labeled with DAPI (blue), proliferating cells were labeled with BrdU antibody (red), and cells whose nuclei were colocalized with BrdU were labeled magenta. (**G**): Comparison of the frequency of proliferating cells in the control and *Myomaker*-siRNA groups. Effect of interfering with *Myomaker* on expressing *Pax7*, *MyoD* and *MyoG* (**H**–**J**). ns. stands for no significant difference (*p* > 0.05). Scale bar is 25 μm.

**Table 1 animals-14-02448-t001:** The Primers for RT-qPCR.

Genes	Forward Primer Sequence (5′-3′)	Reverse Primer Sequence (5′-3′)
*Pax7*	AGCCACAACATGACTTCTCC	GTCCACCGTCTTAATGGAGG
*MyoD*	CAACGACGCCTTTGAGACCCTG	GTCCGAATCCCGCTGTAGTGT
*MyoG*	CGAGACCAACCCTTACTTCTTCCCT	GACTCCCACACAAGCCCATCAT
*Myomaker*	AGTGTTTACGGCACGGCTC	CGTGGTCAACACTCCAAACAT
*Rpl13*	CACAAGAAGGAGAAGGCTCGGGT	TTTGGCTCTCTTGGCACGGAT

**Table 2 animals-14-02448-t002:** Effects of interfering with *Myomaker* on body length and weight of Chinese perch juveniles.

Items	Control (Mean ± SEM)	*Myomaker*-siRNA (Mean ± SEM)
Initial weight (g)	2.31 ± 0.21	2.30 ± 0.24
Final weight (g)	6.40 ± 0.23	4.24 ± 0.28
Initial length (cm)	5.0 ± 0.28	4.9 ± 0.30
Final length (cm)	6.6 ± 0.28	5.9 ± 0.22

## Data Availability

The datasets produced during the study are obtainable from the corresponding researchers upon request.

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
