# Peer review of "The Regulatory Role of Myomaker in the Muscle Growth of the Chinese Perch (Siniperca chuatsi)"

_animals, 2024, doi:10.3390/ani14172448_

Round 1
Reviewer 1 Report
Comments and Suggestions for Authors
Minors
1. Pls use the same words throughout the study such as SE or SEM?
2. Pls add the mean +- SEM in table 2.
3. Other comments are found in the attached files.
Majors
Tittle
I suggested the changing tittle into “The regulatory role of Myomaker in the muscle growth of the Chinese perch (Siniperca chuatsi)”
Introduction
I am sure the important reason why the author is interested in fast muscle (see from the tittle) ?.
Methods
Several points are found (see the attached file). Pls specific locations in the muscle area such as the dorsal (epaxial) and ventral (hypaxial) body muscles.
Results
1. Histological observation (in Figure 4), I suggested that the external morphology and histological features of fast and slow muscles should be recommended. Additionally, the authors are not explained in muscle histological details. Pls add the details as mentioned above.
In Figure 4A-4B compared between control and siRNA. I am sure that these figures are not same with the magnification. Pls use the same magnification between figures.
2. In Figures 4C-4E, all morphometric parameters in the muscle are shown, however, the methods or collecting data (such as Frequency of fiber or fiber diameters) are not found under the standard protocol. Pls add these data in the methodology (in 2.6)
3. In Figure 6, the H and E staning method of muscle should be added and compared with the DAPI figure. It is easy to understand for reader.

Minor editing of English language required
Author Response
Dear Reviewer,
Thank you very much for the reviews on our manuscript. We appreciate your thoughtful and constructive suggestions, which will make our paper significantly improved. We have carefully revised the manuscript according to all the referees’ comments, and the changes were marked in red in the revised paper. Below we repeat your comments, and add our response including what changes have been made to the manuscript accordingly.
Comments
Minors
- Pls use the same words throughout the study such as SE or SEM?
Response: Thank you for bringing this to our attention. The “SE” is a typo. We have revised 'SE' to 'SEM' throughout the manuscript as per your suggestion.
- Pls add the mean±SEM in table 2.
Response: Thank you for your suggestion. We have now included the mean ± SEM in Table 2 of revised manuscript.
- Other comments are found in the attached files.
(1) More explain to details both feeding condition and sampling methods.
Response: Thank you for the suggestions you gave regarding the description of the experimental methods. The Chinese perch were reared in recirculating aquaria with an incubation water temperature of 24 ± 1 °C, a dissolved oxygen level of 8 ± 0.2 mg/L, and a pH range of 7.4-7.7. Feed equal amounts of live Megalobrama amblycephala juveniles twice daily (8:00 am and 6:00 pm). We added the relevant description in 2.1. Experimental animals. A detailed description of the sampling methods was added to 2.3 Tissue sampling and Myomaker-siRNA inhibition assay.
(2) pls specific stage or sex of the sampled fishes?
Response: The Chinese perch for the tissue expression experiments were 5-month-old juveniles, the siRNA interference experiments were performed on 1-month-old juvenile fish, at this stage the myoblast fusion may be active in Chinese perch. Since the sexual maturity of the Chinese perch is about one year, the Chinese perch has no obvious gender characteristics in the juvenile stage, and there is no obvious difference in the growth rate of male and female before 5-month-old, so this experiment did not distinguish the sex of the fish.
(3) Pls specific reasons how to separate between fast and slow muscle?
Response: Fish muscle can be divided into two main types, fast muscle and slow muscle. Fast muscle also known as white muscle, the primary portion of the muscle tissues, is located in the deep layer of the myotome, while slow muscle (red muscle) is a superficial longitudinal band located under the skin on the lateral line (refer to Chen et al., 2023, International Journal of Biological Macromolecules). This provides a clearer distinction between fast and slow muscle.
(4) How? what are the parameters that you are followed?
Response: We have revised the manuscript to explain that the interference times and doses for Myomaker-siRNA were determined based on previously reported studies. We have updated the text to: “The interference times and doses of Myomaker-siRNA were determined based on previous reports.” Thank you for helping us improve the precision of our methods.
(5) I think that the author should be showed the figures to support of this data, as a good and easy to understand.
Response: Thank you for your suggestions, we have added a schematic diagram of the sampling area in the supplementary Fig. S1.
(6) pls changed treated into "fixed".
Response: Thanks to your suggestion, we have changed "treated" to "fixed" in the revised manuscript.
(7) pls delete since the author proposed in the methodology (the first sentence in Result 3.4 and 3.6).
Response: Thank you for your suggestion. We have removed the description of the experimental methodology that was repeated in the results sections 3.4 and 3.6.
Majors
Tittle
I suggested the changing tittle into “The regulatory role of Myomaker in the muscle growth of the Chinese perch (Siniperca chuatsi)”
Response: Thank you for your suggestion, we have updated the title according to your suggestion.
Introduction
I am sure the important reason why the author is interested in fast muscle (see from the tittle)?
Response: We are more interested in fast-twitch muscle because fast muscle is the main component of skeletal muscle in fish, comprising over 80% of the muscle. We have added some sentences to describe fast and slow muscles in fish in the introduction to help readers understand the difference between fast and slow muscles.
Methods
Several points are found (see the attached file). Pls specific locations in the muscle area such as the dorsal (epaxial) and ventral (hypaxial) body muscles.
Response: Thank you for your suggestion, we have added a schematic diagram of the sampling area in the supplementary Fig. S1. We hope that it will help readers to better understand the sampling position.
Results
- Histological observation (in Figure 4), I suggested that the external morphology and histological features of fast and slow muscles should be recommended. Additionally, the authors are not explained in muscle histological details. Pls add the details as mentioned above.
In Figure 4A-4B compared between control and siRNA. I am sure that these figures are not same with the magnification. Pls use the same magnification between figures.
Response: Comparison of external morphological and histological characteristics of fast and slow muscles of Chinese perch is an interesting work. Due to the proportion of slow muscle in the skeletal muscle of Chinese perch is minimal before 4-month-old, and it is difficult to separate it. When the siRNA interference experiment ended, the Chinese perch was nearly 2-month-old, so we only conducted histological analysis on fast muscle in this study. Fig. 4 shows the tissue section of fast muscle, but slow muscle is not included.
We have rechecked the magnification of our manuscript images, the Fig. 4A and Fig. 4B were taken under the same inverted microscope (DMI3000B, Leica, UK), we determined that both images were at the same magnification. As shown in Fig. 4B of the revised manuscript, we have provided a new image with a scale bar that comes with the camera system.
- In Figures 4C-4E, all morphometric parameters in the muscle are shown, however, the methods or collecting data (such as Frequency of fiber or fiber diameters) are not found under the standard protocol. Pls add these data in the methodology (in 2.6)
Response: Thank you for your suggestions regarding Figure 4C-4E. Based on your suggestion, we have carefully revised the Methods section to include detailed descriptions of the methods used for collecting data on muscle fiber diameters in section 2.6.
- In Figure 6, the H and E staning method of muscle should be added and compared with the DAPI figure. It is easy to understand for reader.
Response: Thanks for your helpful comment on Figure 6. At present, we cannot perform immunofluorescence detection and HE staining on the same section at the same time. So, we have adjusted the exposure settings to enable visualization of the boundary of muscle fibers in comparison with the DAPI-stained sections. Please see in Fig. 6.
We have done our best to address your comments in the revision. We hope that you find the changes appropriate, and looking forward to hearing your decision. If there are any comments on the revised version, we will do our best to continue to improve our manuscript.
With best regards,
Sincerely yours,
Wuying Chu
College of Biological and Chemical Engineering, Changsha University, Changsha, Hunan, China 410022
Reviewer 2 Report
Comments and Suggestions for Authors
The manuscript carried out by Zeng et al. aimed to investigate the role of the muscle-specific membrane protein Myomaker in the post-hatching growth and development of skeletal muscle in the economically important Chinese perch (Siniperca chuatsi). Given that Myomaker is known to regulate myoblast fusion, the study sought to determine its specific function in fast muscle tissue. By employing siRNA to knockdown Myomaker expression, authors aimed to assess its impact on muscle growth, muscle fiber diameter, and the number of nuclei within single muscle fibers. The findings enhance our understanding of Myomaker's role in muscle development and growth in Chinese perch, potentially informing strategies to improve aquaculture practices for this species. Generally speaking, the manuscript is well designed and organized. However, I have some comments for the authors to address.
Comments:
1. Which month did you treat the fish?
2. Why did you select a 2 mg/kg concentration? Have you tested the higher concentrations?
3. Do you think factors such as season, diet, nutrition, temperature, or other environmental conditions affected the performance of your experiment?
4. Some grammar and format mistakes were found. Please check carefully. I have listed some of them below.
Minor points:
1. Line 4, “Wuying Chu a,*” remove (,)
2. Lines 15,56, Italicize the word “Mayomaker” and consider double-checking throughout the manuscript.
3. In lines 47-51, to avoid repetition, remove the phrase “in myoblast fusion” from the sentence that starts with “The first……”
4. Line 76-77, Rephrase “Chinese perch is now preferred by more and more consumers, and the demand for Chinese perch in the market has been increasing” to “Chinese perch is increasingly preferred by consumers, leading to a rising market demand”
5. Line 79, define “siRNA”
6. Line 97-101, please rephrase and doublecheck for common errors
7. Line 107, please provide the actual average size and weight of the selected Chinese perch
8. Line 111, what were the criteria used to choose 20 µM of siRNA to inject the fish in the siRNA group every 7 days for 21 days?
9. Line 161, “(Shi et al., 2018) “ please use standard citation style for this journal
10. Line 269, replace “n.s” with “ns”
Comments on the Quality of English LanguageMinor editing of English language required.
Author Response
Dear Reviewer,
Thank you very much for the reviews on our manuscript. We are glad to see you supportive of our research, and we appreciate your thoughtful and constructive suggestions, which will make our paper significantly improved. We have carefully revised the manuscript according to all the referees’ comments, and the changes were marked in red in the revised paper. Below we repeat your comments, and add our response including what changes have been made to the manuscript accordingly.
Comments:
- Which month did you treat the fish?
Response: We conducted the siRNA interference experiments in June, 2023. Specifically, the Chinese perch selected in our experiment was hatched in early May last year. Because we chose 1-month-old Chinese perch for processing, the experiment was carried out in June.
- Why did you select a 2 mg/kg concentration? Have you tested the higher concentrations?
Response: We chose 2 mg/kg for injection based on the manufacturer's guidelines and previous studies. In preliminary experiment, we found that 2 mg/kg significantly inhibited Myomaker expression. This result informed our decision to use this specific dosage for subsequent experiments.
- Do you think factors such as season, diet, nutrition, temperature, or other environmental conditions affected the performance of your experiment?
Response: Thank you for your thoughtful consideration regarding potential environmental factors that may influence our experimental outcomes. We acknowledge that factors such as season, diet, nutrition, temperature, and other environmental conditions can indeed impact experimental results. To minimize these potential effects, all experiments were conducted indoors under controlled conditions. Both the control and experimental groups were maintained under identical and appropriate dietary, temperature, and environmental conditions throughout the study period. This approach was taken to ensure the reliability and reproducibility of our experimental results.
- Some grammar and format mistakes were found. Please check carefully. I have listed some of them below.
Minor points:
- Line 4, “Wuying Chu a, *” remove (,)
Response: Thank you for your suggestion. We have now made the necessary correction as per your recommendation.
- Lines 15,56, Italicize the word “Myomaker” and consider double-checking throughout the manuscript.
Response: Thank you for your suggestion. In this manuscript, gene names were used for protein in normal and mRNA in italics. In lines 15, 56, where italics are not used because "Myomaker" denotes protein and therefore is not italicized, we have double-checked throughout the manuscript.
- In lines 47-51, to avoid repetition, remove the phrase “in myoblast fusion” from the sentence that starts with “The first……”
Response: Thanks to your suggestion. we have removed the duplicate "in myoblast fusion".
- Line 76-77, Rephrase “Chinese perch is now preferred by more and more consumers, and the demand for Chinese perch in the market has been increasing “to “Chinese perch is increasingly preferred by consumers, leading to a rising market demand”
Response: Thank you for your valuable feedback regarding the manuscript. We have revised the sentence as suggested.
- Line 79, define “siRNA”
Response: Thanks to your suggestion, we have defined the siRNA targeting Myomaker, including the text and figures.
- Line 97-101, please rephrase and double check for common errors
Response: We have defined the siRNA targeting Myomaker, including the text and figures.
- Line 107, please provide the actual average size and weight of the selected Chinese perch
Response: The Chinese perch used for tissue expression experiments weighed 210 ± 10 g, and the Chinese perch used for siRNA interference experiments weighed 2.3 ± 0.3 g, these data was provided in 2.1 Experimental animals.
- Line 111, what were the criteria used to choose 20 µM of siRNA to inject the fish in the siRNA group every 7 days for 21 days?
Response: 20 µM siRNA concentration was chosen based on the supplier's recommendation, while the injection times were adapted from the experimental protocol for siRNA interference with MSTN by Khan et al., 2016. After 21 days of siRNA interference, the expression level of MSTN was significantly decreased, affecting the muscle mass, size, and strength of the animals.
- Line 161, “(Shi et al., 2018) “please use standard citation style for this journal
Response: Thanks to your reminder, we have modified the citation format.
- Line 269, replace “n.s” with “ns”
Response: We have changed “n.s” to “ns”.
We have done our best to address your comments in the revision. We hope that you find the changes appropriate, and looking forward to hearing your decision. If there are any comments on the revised version, we will do our best to continue to improve our manuscript.
With best regards,
Sincerely yours,
Wuying Chu
College of Biological and Chemical Engineering, Changsha University, Changsha, Hunan, China 410022
Reviewer 3 Report
Comments and Suggestions for Authors
I checked your manuscript and described comments below.
Chinese perch (Siniperca chuatsi) is an important ingredient in luxury food in China.
Fish skeletal muscle is related to taste, so it is important to study about muscle.
There have already been several research papers on the skeletal muscles of the Chinese perch (Siniperca chuatsi).
However, there have been no papers yet on the use of siRNA to control the expression of skeletal muscles in Chinese perch (Siniperca chuatsi), and I believe this paper is significant in this regard.
I think you should consider the following points.
1. About Figure 1-A, DUF3522 is a transmembrane domain of unknown function. It is probably shown in Figure 1-B, but I think it would be better to show the transmembrane part of this domain in this figure.
2. As for Figure 1-B, I think it would be better to include the results of TMHMM for Danio rerio, NP_001002088.1; Micropterus salmoides, XP_038559766.1; Meg-192 alobrama amblycephala, XP_048045299.1; Sander lucioperca, 193 XP_031161515.1; Xenopus laevis, XP_041428255.1; Mus musculus, NP_079652.1; and Homo sapiens, 194 NP_001073952.1 in addition to Siniperca chuatsi, XP_044052078.1.
3. It might be better to state that myomaker is myoblast fusion factor (mymk).
I don't think this paper has major problems and grammatical problems.
Author Response
Dear Reviewer,
Thank you very much for the reviews on our manuscript. We are glad to see you supportive of our research, and we appreciate your thoughtful and constructive suggestions, which will make our paper significantly improved. We have carefully revised the manuscript according to all the referees’ comments, and the changes were marked in red in the revised paper. Below we repeat your comments, and add our response including what changes have been made to the manuscript accordingly.
Comments:
- About Figure 1-A, DUF3522 is a transmembrane domain of unknown function. It is probably shown in Figure 1-B, but I think it would be better to show the transmembrane part of this domain in this figure.
Response: Thanks to your suggestion, we have marked the transmembrane domain of the Chinese perch in Figure 1-A.
- As for Figure 1-B, I think it would be better to include the results of TMHMM for Danio rerio, NP_001002088.1; Micropterus salmoides, XP_038559766.1; Meg alobrama amblycephala, XP_048045299.1; Sander lucioperca, XP_031161515.1; Xenopus laevis, XP_041428255.1; Mus musculus, NP_079652.1; and Homo sapiens, NP_001073952.1 in addition to Siniperca chuatsi, XP_044052078.1.
Response: We appreciate your recommendation, we have used TMHMM2.0 to re-analyze the transmembrane domains for the additional species you mentioned. We include these results in Figure 1 to provide a more comprehensive comparison.
- It might be better to state that Myomaker is myoblast fusion factor (mymk).
Response: Thanks for your suggestion. We updated the Abstract and Introduction of the manuscript to show that Myomaker is the myoblast fusion factor.
We have done our best to address your comments in the revision. We hope that you find the changes appropriate, and looking forward to hearing your decision. If there are any comments on the revised version, we will do our best to continue to improve our manuscript.
With best regards,
Sincerely yours,
Wuying Chu
College of Biological and Chemical Engineering, Changsha University, Changsha, Hunan, China 410022
Round 2
Reviewer 1 Report
Comments and Suggestions for Authors
All comments are cleared and this manuscript is already published.
This data from the manuscript will be exclusively considered to be a good publication.
Comments on the Quality of English LanguageMinor editing of English language required.
Reviewer 2 Report
Comments and Suggestions for Authors
It is ok for acceptance.